## [Transparent Peer Review File · Nature Communications]

Transient molecular chimerism for exploiting xenogeneic organelles

Corresponding Author: Professor Yuichiro Kashiyama

Version 0:

Reviewer comments:

Reviewer #1

(Remarks to the Author)

The authors present exiting work on *Rapaza viridis*, a unique marine euglenid that retains chloroplasts only by constant replenishment of their organellar complement through kleptoplasty. The biology of this system is in itself thrilling.

Commendably, Kashiyama and colleagues have taken the investigation into this system to the next level by presenting genetic studies on the joint action of the genetic compartments -- host nucleus and kleptoplast -- to maintain kleptoplast function in an essential process.

This is outstanding and the work is of extremely high quality. I just have a few remarks.

The interaction between plastid and nucleus is often considered as one of the outstanding features of streptophytes, especially land plants; see e.g.:

Goldbecker, E. S. & de Vries, J. Systems Biology of Streptophyte Cell Evolution. Annual Review of Plant Biology 76, 493-522 (2025). <https://doi.org/10.1146/annurev-arplant-083123-060254>

There are also some avenues in chlorophytes:

Duanmu, D. et al. Retrograde bilin signaling enables *Chlamydomonas* greening and phototrophic survival. Proc Natl Acad Sci U S A 110, 3621-3626 (2013). <https://doi.org/10.1073/pnas.1222375110>

I think unpacking this, summarizing what we know in streptophyte and chlorophytes in light of their evolution, and how the needs of the kleptoplasts might be communicated to the host in this unique setting would make a very interesting additional discussion point.

Adding a bit of a summary of which key genes are encoded in which compartment (e.g. to Figure 1) would help a lot making this work more easily accessible.

Reviewer #2

(Remarks to the Author)

First, I would like to congratulate the authors on this very comprehensive and elegant piece of work that provides important insights into the biology of kleptoplasts, providing clear evidence for the import of host-encoded proteins. The work beautifully demonstrates that integration of kleptoplasts can evolve beyond the transient survival of the "stolen plastids" supported solely by the proteins they bring along and highlights kleptoplasts as a possible intermediate on the route to acquisition of a novel complex plastid that is permanently retained in the "host" species.

Despite my overall enthusiasm for the work, I have a number of issues (mostly minor) that need to be addressed before I can recommend accepting the manuscript for publication.

Major issues

- Fig. 3 c-e and Fig. 4e: Coomassie staining is generally considered incompatible with downstream Western blotting as it fixes the proteins in the gel. Nevertheless, the authors chose Coomassie staining of the gel as a loading control for Western blotting (Extended Data Fig. 1). The Methods section on Immunoblotting does not mention a Coomassie staining step. Thus, it is unclear what is shown in Extended Data Fig. 1. Are these the gels before transfer using a Western blot-compatible Coomassie staining protocol? If so, details have to be provided in the methods. Or are these parallelly loaded SDS-PAGE gels with the same samples. If so, this is not a suitable control for equal loading. Either a staining compatible with downstream Western blot (such as TCE staining) or immune detection of a house-keeping protein on the same membrane is required. Since protein amounts are critical for the interpretation of the experiments (especially in Fig. 4e), proper loading

controls are essential here.

- Lines 112-117: "Phylogenetic analysis of the translated peptide sequences showed that [...] about 20% [clustered] with *Tetraselmis* spp.. However, none were identical or highly homologous to the current kleptoplast donor strain *Tetraselmis* sp. NIES-4478. This suggests any horizontal transfers of these important genes occurred in the distant past and the genes evolved as unique components of *R. viridis*." This statement seems to be in contradiction with the information that de novo transcriptome sequences were filtered against the genome data from *Tetraselmis* sp. and contigs with 97% identity were omitted" (lines 579-584). -> Since contigs identical or very similar to *Tetraselmis* were actively removed following transcriptome assembly, it is not surprising that such transcripts are missing in the final dataset. An important question that should be addressed to answer the question if recent gene transfers occurred would be if any of the transcripts identical or very similar to *Tetraselmis* contain the host-specific SL at their 5' end.
- An interesting question that is not addressed by the study concerns the extent of protein import into the kleptoplasts. Can the identified features in the different classes of targeting signals be used to bioinformatically predict further kleptoplast-targeted proteins? Please explain why not or include this analysis in the revised version of the manuscript.

Minor issues

- Title and throughout the text: The expression "organelloid" is uncommon and appears inapt to describe a kleptoplast. The term "organoid" (meaning organ-like) is commonly used for synthetic miniature models of organs that replace the actual organs in research. A kleptoplast, however, would rather resemble a transplanted organ, not a synthetic miniature replica. Therefore, I would recommend that the authors stay with the established term kleptoplast and specify that this kleptoplast is transiently operated with the help of nucleus-encoded kleptoplast-targeted proteins.
- Line 95: "insufficient to resolve the intricacies of these unique systems" -> This is a rather vague statement. Please explain precisely for what the predictions are insufficient for? E.g., insufficient to prove protein import and unravel its mechanistic underpinnings.
- Line 105-108: These transcripts were homologous to those found in the chloroplast-targeted proteins of photosynthetic eukaryotes based on the time-series transcriptome data (Supplementary Data) from four distinct kleptoplastic stages (Fig. 1), whose expression levels were comparable to key mitochondrial metabolic genes (Extended Data Table 1, Fig. 2a). -> Awkward, long sentence. I can only guess its meaning. Please rephrase.
- Line 167: Replace "Fig. 3h" and "Fig. 3i" by "Fig. 3g"
- Lines 217-219: "Based on the results suggesting that the NtLCDs of RvRbcS-like and RvRca-like act as translocation signals and are cleaved upon maturation (Fig. 3a, Extended Data Table 2)" -> Incomplete sentence and unclear how Extended Data Table 2 supports this statement. Correct citation?
- Line 226: Change reference to Extended Data Table 2 to Extended Data Table 3.
- Line 251: Term "structural recruitment" is not clear.
- Line 307: Expression "permanent transient kleptoplasts" is contradictory in itself. Please rephrase.
- Lines 341-344: "*R. viridis* remains highly photosynthetic for approximately 2 weeks, remaining virtually autotrophic through a 1-week growth phase and a 1-week stationary phase. This is supported by sustained growth without external inorganic nitrogen⁹, culture decline in the absence of light⁸, and cytosolic polysaccharide accumulation in stationary phase cells^{8,9}." -> It is unclear why independence of inorganic nitrogen supports high photosynthetic activity or autotrophy. Please rephrase or explain.
- Line 621: Please provide database name along with accession numbers.
- Lines 660-668: The description of the RNAi procedure is not very clear. Please rephrase the whole section. Are 15 µg dsRNA introduced in total or 15 µg per electroporation? When was the electroporation procedure repeated? 2 d after the first round? When was the experiment evaluated? 3 d after the second electroporation round? I have to guess these parameters. They are not very clear from the text. What is the source of the seawater? Does 5% seawater mean 5% (v/v) seawater in dH₂O or does 5% indicate the salinity of the seawater?
- Line 672: Maruyama et al is not reference 8. Please correct.
- Line 681: ...using previously described electroporation conditions⁸. Please check if 8 is the correct reference or if it should be 9.
- Lines 670-688: For the CRISPR/Cas9 genome editing section, information on how cultures were handled following electroporation is missing. Please include a detailed description of procedures and timing between electroporation and analysis of the generated cells.
- Fig. 1: What are the large light grey structures in cells in the declining state?
- Fig. 3 g: I assume, the labelling of RvRca and Wt are inverted.
- Fig. 4d: Scale bars are required for the microscopic images shown.
- Fig. 6: Cys is listed as polar as well as non-polar amino acid. The numbers in the figure legend don't fit to the main text and the proteins shown in the figure. Legend: "Of 29 sequences examined, 27 exhibited extended NtLCDs". Figure and main text: 37 proteins analyzed in total, 35 with classifiable NtLCDs. Furthermore, the actual alignment underlying the NtLCD analysis shown in Fig. 6 should be provided in the supplemental material.
- Extended Data Table 3: Define "unread gap".
- Extended Data Table 4: Definition required for the numbers provided in brackets behind the "Yes" and "Possible" entries under "Phylogenetic analysis" (bootstrap support, I assume(?)).
- Extended Data Fig. 4: Legend doesn't fit to figure. The described panel a is missing. What are orange and blue in panel b?

Reviewer #3

(Remarks to the Author)

This study by Kashiyama et al. is logical and technically sound. Controls are properly employed, and the results support the main experimental claims. The main contribution of this study is the experimental confirmation of the targeting and translocation of two functionally important kleptoplast-localized nucleus-encoded proteins in *Rapazas viridis*. This is an important technical advance and is much welcomed.

From a biological and physiological perspective, neither the questions nor the results are novel or unexpected. The proteins chosen are expected to be targeted to the kleptoplasts given an abundance of circumstantial evidence provided in previous studies. The study is technically impressive, but the technical advancements (i.e., knock-down, CRISPR-Cas9) had previously been developed and reported by Maruyama et al.

In my opinion, the framing of the manuscript is misleading. The title is rather cryptic and overhyped. These concerns are mostly restricted to the title and the introduction. Unnecessary neologism such as "xenogeneic organelloids" are introduced to presumably exaggerate the importance of this contribution.

The statement in L319-323 is somewhat confusing and not well-informed based on evolutionary theory. Extant living species cannot be considered evolutionary intermediates, but there are characters that can be considered as such. Species express combination of characters that can be considered ancestral or derived. In this way, *R. viridis* is informative as it retains systems that may have been ancestral to photosynthetic euglenids and may therefore inform the evolutionary origin of phototrophy in this group.

Version 1:

Reviewer comments:

Reviewer #1

(Remarks to the Author)

The authors have addressed all my remarks to my fullest satisfaction. I do not have any further comments and congratulate them on a landmark paper.

Reviewer #2

(Remarks to the Author)

The revised version of the manuscript has improved substantially, and the authors are to be congratulated on the new version. Almost all of this reviewer's previous criticisms have been addressed sufficiently. However, the terms "5% seawater" and "10% seawater" remain unclear and should be specified more precisely as requested previously.

Reviewer #3

(Remarks to the Author)

I stand by my previous comments, and I am not persuaded by the author's responses. I leave it to the editor to make a decision based on the significance and novelty of the paper for the journal.

RESPONSES TO REVIEWER COMMENTS

Reviewer #1:

The authors present exiting work on *Rapaza viridis*, a unique marine euglenid that retains chloroplasts only by constant replenishment of their organellar complement through kleptoplasty. The biology of this system is in itself thrilling. Commendably, Kashiyama and colleagues have taken the investigation into this system to the next level by presenting genetic studies on the joint action of the genetic compartments -- host nucleus and kleptoplast -- to maintain kleptoplast function in an essential process.

We are honored by your kind words. We are pleased that you have taken an interest in this study.

This is outstanding and the work is of extremely high quality. I just have a few remarks.

- The interaction between plastid and nucleus is often considered as one of the outstanding features of streptophytes, especially land plants; see e.g.:

Goldbecker, E. S. & de Vries, J. Systems Biology of Streptophyte Cell Evolution. *Annual Review of Plant Biology* **76**, 493-522 (2025). <https://doi.org/10.1146/annurev-arplant-083123-060254>

There are also some avenues in chlorophytes:

Duanmu, D. et al. Retrograde bilin signaling enables *Chlamydomonas* greening and phototrophic survival. *Proc Natl Acad Sci U S A* **110**, 3621-3626 (2013). <https://doi.org/10.1073/pnas.1222375110>

I think unpacking this, summarizing what we know in streptophyte and chlorophytes in light of their evolution, and how the needs of the kleptoplasts might be communicated to the host in this unique setting would make a very interesting additional discussion point.

Your comment was refreshing and drew attention to aspects we had overlooked. The papers you recommended, especially the first one, offer stimulating reviews with important implications for evolutionary thinking.

In fact, most kleptoplasty studies, including ours, have mainly focused on direct structural control by nucleus-encoded proteins acting on kleptoplasts but not much about communications. Regarding *Rapaza viridis*, two issues would be central: (1) anterograde control by the nucleus over transcription and translation of the kleptoplast genome and (2) whether retrograde signals from the chloroplast are actively received and utilized by the nucleus, potentially through factors in the *R. viridis* cytosol or the outermost envelope (i.e., the phagosomal membrane). Such control is crucial for permanent integration, particularly in the presence of reactive oxygen species.

However, we consider it premature to address these topics here. First, our comparative transcriptomics were not designed around light–dark cycles that critically affect photosynthetic physiology. All samples were collected during the light phase in order to *exclude* diurnal effects but to isolate kleptoplast maturation. Thus, our data are not suited to detect diurnally varying nucleus-chloroplast communications. Second, *Tetraselmis* (Chlorodendrophyceae) is phylogenetically distant from the commonly used chlorophyte model, *Chlamydomonas* (Chlorophyceae). Simple extrapolation of retrograde signaling of the chlorophyte would not be very appropriate. For example, *Tetraselmis* possesses phytochromes, analogous to those found in streptophytes, and therefore likely employs signaling pathways distinct from those of chlorophytes.

Additionally, comprehensive reviews of Chlorophyta (other than Chlorophyceae) are lacking. Third, we did not detect indicative factors of such a communication among the highly expressed genes we analyzed. At moderate expression levels, however, we find suggestive candidates, such as a putative kleptoplast-targeted heme oxygenase and sequences compatible with chloroplast σ factors. Selectively including these candidates would, however, blur the focus of the current manuscript.

For these reasons, we have decided to only briefly touch on bidirectional communication between the kleptoplast and the *R. viridis* nucleus as a topic for future research. Accordingly, we have inserted the following sentence into the latter half of the final paragraph (P14-15; L324-327) as a proposal for future research.

Beyond the direct structural manipulation by host-supplied proteins, the interaction may involve precise anterograde and retrograde signalling between the host nucleus and the organelle⁵¹⁻⁵³, which would be essential for regulating photosynthesis to prevent the xenogenic object from merely becoming a source of reactive oxygen species.

- Adding a bit of a summary of which key genes are encoded in which compartment (e.g. to Figure 1) would help a lot making this work more easily accessible.

We found this to be a very good suggestion. We have now added a Venn diagram showing the genes encoding subunits of key photosynthetic complexes in Fig. 1e (P16).

Reviewer #2 (Remarks to the Author):

First, I would like to congratulate the authors on this very comprehensive and elegant piece of work that provides important insights into the biology of kleptoplasts, providing clear evidence for the import of host-encoded proteins. The work beautifully demonstrates that integration of kleptoplasts can evolve beyond the transient survival of the “stolen plastids” supported solely by the proteins they bring along and highlights kleptoplasts as a possible intermediate on the route to acquisition of a novel complex plastid that is permanently retained in the “host” species.

Despite my overall enthusiasm for the work, I have a number of issues (mostly minor) that need to be addressed before I can recommend accepting the manuscript for publication.

Major issues

- Fig. 3 c-e and Fig. 4e: Coomassie staining is generally considered incompatible with downstream Western blotting as it fixes the proteins in the gel. Nevertheless, the authors chose Coomassie staining of the gel as a loading control for Western blotting (Extended Data Fig. 1). The Methods section on Immunoblotting does not mention a Coomassie staining step. Thus, it is unclear what is shown in Extended Data Fig. 1. Are these the gels before transfer using a Western blot-compatible Coomassie staining protocol? If so, details have to be provided in the methods. Or are these parallelly loaded SDS-PAGE gels with the same samples. If so, this is not a suitable control for equal loading. Either a staining compatible with downstream Western blot (such as TCE staining) or immune detection of a house-keeping protein on the same membrane is required. Since protein amounts are critical for the interpretation of the experiments (especially in Fig. 4e), proper loading controls are essential here.

These in Extended Data Fig. 1 were Coomassie-stained SDS-PAGE gels that were loaded in parallel using the same volumes of identical cell lysates for each sample and electrophoresed simultaneously in the same chamber. We understand that equal loading should be particularly critical for the Western blot analyses shown in Fig. 4e and thus decided to take your comment seriously there. Following your suggestion, we prepared another set of parallel time-course samples and repeated the entire experiment. This included additional immunoblotting using an anti- α -tubulin antibody to verify protein loading in each lane.

Because our samples contain two distinct organisms, the alga *Tetraselmis* and the flagellate *R. viridis*, as well as representing different cellular stages of the latter, we think the minor variation observed among the tubulin signal observed across the series was an understandable consequence. Indeed, this occurred even though we estimated equal protein masses from each sample tube. Importantly, the new results with the α -tubulin signals reinforce our previous interpretation. In particular, in the $\Delta RvRbcS$ -like time series samples, the *TsRbcS* and *RbcL* signals decreased on days 4 and 7, while the α -tubulin signal increased slightly.

Accordingly, we replaced Fig. 4e with the new data (P19). In addition, we conducted two more immunoblotting experiments as independent triplicates, starting from separate culture batches, and obtained densitometry data ($n = 3$), including the blot shown in Fig. 4e. Using these data, we updated Fig. 4f. (P19). During this process, we discovered an error in the previous Excel-based calculations, which had inadvertently exaggerated the apparent decrease in *TsRbcS* levels. The newly generated plots, prepared with greater care, support what we consider to be the appropriate trend. In the new Fig. 4f, we also present the time-series changes in *RbcL*. Corresponding revisions have been made to the main text (P9; L195–202), as well as to the figure caption and Methods section.

- Lines 112-117: “Phylogenetic analysis of the translated peptide sequences showed that [...] about 20% [clustered] with *Tetraselmis* spp.. However, none were identical or highly homologous to the current kleptoplast donor strain *Tetraselmis* sp. NIES-4478. This suggests any horizontal transfers of these important genes occurred in the distant past and the genes evolved as unique components of *R. viridis*.” This statement seems to be in contradiction with the information that de novo transcriptome sequences were filtered against the genome data from *Tetraselmis* sp. and contigs with 97% identity were omitted” (lines 579-584). -> Since contigs identical or very similar to *Tetraselmis* were actively removed following transcriptome assembly, it is not surprising that such transcripts are missing in the final dataset. An important question that should be addressed to answer the question if recent gene transfers occurred would be if any of the transcripts identical or very similar to *Tetraselmis* contain the host-specific SL at their 5' end.

First, we discussed in the paragraph you mentioned regarding exclusively on the 37 highly expressed genes. So, what we meant was that none of these apparently important genes are of recent horizontal gene transfer from the donor *Tetraselmis*. Thus, we slightly modified the phrase of the main text to clarify our intent (P5; L113).

Moreover, following your comment, we actually examined whether any sequences matching *Tetraselmis* at over 97% similarity (i.e., those excluded from the reference dataset) contained Euglenid-type spliced leader sequences at their ends. We found none. This information has now been added to the text in Methods section.

- An interesting question that is not addressed by the study concerns the extent of protein import into the kleptoplasts. Can the identified features in the different classes of targeting signals be used to bioinformatically

predict further kleptoplast-targeted proteins? Please explain why not or include this analysis in the revised version of the manuscript.

In our earlier paper (Karnkowska et al., 2023, *PNAS*), we extrapolated the features of the *Euglena gracilis* chloroplast-targeting signal to search for sequences in *Rapaza viridis* that might be directed to kleptoplasts and discussed these candidates. At that stage, however, kleptoplast targeting remained purely hypothetical: there was no experimental evidence and no quantitative transcript data. In the present study, therefore, we conducted biochemical experiments supported by genetic manipulations to demonstrate that at least two of these candidates are indeed transported to kleptoplasts. Importantly, the candidates in *this* study were not selected simply by checking for “signal-like” motifs. Instead, we first identified chloroplast-related genes that are highly expressed in *R. viridis* despite its lack of a bona fide chloroplast (the kleptoplasty itself having been confirmed in the above paper), and then biochemically verified that their products are actually transported. We subsequently analyzed in detail the N-termini of other highly expressed chloroplast-related genes (a point not examined in depth in our *previous* study) and found that, in the majority of cases, these sequences share features with, although are not identical to, the *E. gracilis* chloroplast-targeting signal. Thus, this study represents the first deductive approach to identify and evaluate kleptoplast-targeting signals.

While we agree that a more comprehensive and higher-precision bioinformatic search for kleptoplast-targeting signals beyond our 2023 study may become feasible in the future, we consider it premature to make strong claims at this stage. One reason is that no concrete information is yet available regarding the translocation machinery. Speculation based solely on sequence similarity is undesirable in the present situation, and we should bear in mind that the mechanism remains unresolved even in *E. gracilis*. Moreover, in 2023 we necessarily adopted an empirical strategy for identifying *E. gracilis* chloroplast-targeted proteins; that is, by combining an existing signal-peptide prediction model (notably, only the older version worked adequately, while newer versions did not) with an existing transit-peptide prediction model, both of which were exclusively designed for green plants. In contrast, the present study enables us to demonstrate kleptoplast-targeting sequence function deductively. We therefore consider the next logical step to be a comprehensive analysis of the proteins that are actually transported and the characterization of their N-terminal architectures, which may ultimately provide a more reliable, empirically grounded prediction model, even without a full understanding of the underlying machinery.

Gene-by-gene validation using genetic manipulations, as performed here, would be quite challenging, especially for lower-expression candidates. We are therefore currently isolating and purifying mature kleptoplasts for proteomic analyses to conclusively distinguish the *R. viridis*-derived proteins localized inside the kleptoplast from those that are not transported into it. We believe these analyses will provide much more reliable bioinformatic predictions. However, these ongoing results are beyond the scope of the present manuscript, and that is precisely why we wish to refrain from discussing bioinformatic predictions at this time.

Minor issues

- Title and throughout the text: The expression “organelloid” is uncommon and appears inapt to describe a kleptoplast. The term “organoid” (meaning organ-like) is commonly used for synthetic miniature models of organs that replace the actual organs in research. A kleptoplast, however, would rather resemble a transplanted organ, not a synthetic miniature replica. Therefore, I would recommend that the authors stay with the established term kleptoplast and specify that this kleptoplast is transiently operated with the help of nucleus-encoded kleptoplast-targeted proteins.

There were two reasons for introducing the term “organelloid” instead of simply using “kleptoplast.” First, we anticipate that the category of biological phenomena termed “kleptoplasty” will need to be reevaluated in the near future. Phenomena traditionally referred to as kleptoplasty are rather heterogeneous, ranging from systems that transiently maintain only chloroplasts with an uncertain dependence on photosynthesis to systems that retain the entire algal cytoplasm. Adding *R. viridis* to this category increases its cell biological diversity even further. Most importantly, it is highly questionable whether these phenomena are evolutionarily homologous or even lie on the same evolutionary spectrum.

Second, we believe that the importance of *R. viridis* as a new experimental model lies in elucidating the underlying (and likely universal) cellular mechanisms that enabled this phenomenon to evolve. For this reason, we were keen to avoid using the term “kleptoplast” in the title, as it might suggest to readers that it is a special or idiosyncratic phenomenon.

However, we understand your point: the term "organelloid" may also evoke something artificial. Furthermore, our frequent use of “kleptoplast(y)” in the main text makes our terminology self-contradictory. We propose replacing “xenogeneic organelloid” with “xenogeneic organelle” wherever it appears (including in the title). We believe this term is not problematic and remains consistent with our intention to frame the phenomenon within the broader context of phagocytic interactions, as suggested in the Discussion.

- Line 95: “insufficient to resolve the intricacies of these unique systems” -> This is a rather vague statement. Please explain precisely for what the predictions are insufficient for? E.g., insufficient to prove protein import and unravel its mechanistic underpinnings.

Following your suggestion, we have revised the wording to be more specific as shown below (P4-5: L94-96).

However, *in silico* predictions alone are insufficient to prove protein import and elucidate its mechanistic underpinnings, highlighting the need for cellular-level biochemical investigations⁹.

- Line 105-108: These transcripts were homologous to those found in the chloroplast-targeted proteins of photosynthetic eukaryotes based on the time-series transcriptome data (Supplementary Data) from four distinct kleptoplastic stages (Fig. 1), whose expression levels were comparable to key mitochondrial metabolic genes (Extended Data Table 1, Fig. 2a). -> Awkward, long sentence. I can only guess its meaning. Please rephrase.

Together with the previous sentence, we rephrased as follows (P5: L104-107).

From the time-series transcriptome data taken from four distinct kleptoplastic stages (Fig. 1; Supplementary Data), we identified 37 *R. viridis* transcripts encoding proteins with conserved domains indicative of chloroplastic functions³⁷. The expression levels of these transcripts were comparable to those of key mitochondrial metabolic genes (Extended Data Table 1; Fig. 2a).

- Line 167: Replace “Fig. 3h” and “Fig. 3i” by “Fig. 3g”

We corrected as you pointed out (P8: L166).

- Lines 217-219: “Based on the results suggesting that the NtLCDs of *RvRbcS*-like and *RvRca*-like act as translocation signals and are cleaved upon maturation (Fig. 3a, Extended Data Table 2)” -> Incomplete sentence and unclear how Extended Data Table 2 supports this statement. Correct citation?

That was indeed an incorrect citation; it should be “(Fig. 3c, Extended Data Fig. 2)” instead. In addition, we have made minor revisions to the wording to clarify the intent of the sentence (P10: L215-217).

- Line 226: Change reference to Extended Data Table 2 to Extended Data Table 3.

Corrected (P10: L224).

- Line 251: Term “structural recruitment” is not clear.

Rephrased as “structural incorporation” (P11: L249).

- Line 307: Expression “permanent transient kleptoplasts” is contradictory in itself. Please rephrase.

Removed “permanent” (P14: L305). It was an error.

- Lines 341-344: “*R. viridis* remains highly photosynthetic for approximately 2 weeks, remaining virtually autotrophic through a 1-week growth phase and a 1-week stationary phase. This is supported by sustained growth without external inorganic nitrogen⁹, culture decline in the absence of light⁸, and cytosolic polysaccharide accumulation in stationary phase cells^{8,9}.” -> It is unclear why independence of inorganic nitrogen supports high photosynthetic activity or autotrophy. Please rephrase or explain.

It was another error; “sustained” should have been “suspended”. To make this clearer, we have rephrased the sentence as follows (P16: L343-345):

This autotrophy is supported by suspended growth without external inorganic nitrogen⁹, culture decline in the absence of light⁸, and cytosolic polysaccharide accumulation in stationary phase cells^{8,9}.

- Line 621: Please provide database name along with accession numbers.

These sequences are from the DDBJ database, and we have added this information to the manuscript (P31: L630).

- Lines 660-668: The description of the RNAi procedure is not very clear. Please rephrase the whole section. Are 15 µg dsRNA introduced in total or 15 µg per electroporation? When was the electroporation procedure repeated? 2 d after the first round? When was the experiment evaluated? 3 d after the second electroporation round? I have to guess these parameters. They are not very clear from the text. What is the source of the seawater? Does 5% seawater mean 5% (v/v) seawater in dH₂O or does 5% indicate the salinity of the seawater?

We have rewritten the entire “RNAi” section. Please refer to the revised manuscript (P32-33: L664-679). They must be clear now. Also, we clarified the timing of the observations after these electroporation experiments in the caption of Extended Data Fig. 4.

- Line 672: Maruyama et al is not reference 8. Please correct.

Corrected to 9 (P33; L666).

- Line 681: ...using previously described electroporation conditions⁸. Please check if 8 is the correct reference or if it should be 9.

Corrected to 9 (P33; L682).

- Lines 670-688: For the CRISPR/Cas9 genome editing section, information on how cultures were handled following electroporation is missing. Please include a detailed description of procedures and timing between electroporation and analysis of the generated cells.

We have rewritten the entire “CRISPR/Cas9 genome editing” section (P33-34; L680-708). Please refer to the revised manuscript. In this case, however, the “timing between electroporation” is not relevant, unlike in the RNAi experiments, because the genotypes of the edited clones are permanently modified.

- Fig. 1: What are the large light grey structures in cells in the declining state?

They are vacuoles. To make this clearer, we have modified the caption (P16; L345-347) as follows:

d, In the absence of new kleptoplast acquisition, vacuoles begin to form after 3 weeks and gradually expand to occupy most of the cell. The cells ultimately die after 4–5 weeks (declining stage).

- Fig. 3 g: I assume, the labelling of $\Delta RvRca$ and Wt are inverted.

It was indeed an error (inverted). We corrected in the revised figure (P18).

- Fig. 4d: Scale bars are required for the microscopic images shown.

We added a scale bar to the lower right panel and included the scale information in the figure caption (P19).

- Fig. 6: Cys is listed as polar as well as non-polar amino acid. The numbers in the figure legend don't fit to the main text and the proteins shown in the figure. Legend: "Of 29 sequences examined, 27 exhibited extended NtLCDs". Figure and main text: 37 proteins analyzed in total, 35 with classifiable NtLCDs. Furthermore, the actual alignment underlying the NtLCD analysis shown in Fig. 6 should be provided in the supplemental material.

First, Cys was treated as a polar amino acid in the calculation, and the caption has been corrected accordingly (P21; L426). The numerical values in the figure legend were erroneous and have now been revised (P21; L422-423). The actual alignments underlying the NtLCD analyses in Fig. 6b and 6c are now provided in the Supplementary Data⁶⁵. The availability of the raw data has also been noted in the figure caption (P21; L431-432).

- Extended Data Table 3: Define "unread gap".

We have added text explaining the definition of "unread gap" in the caption of Extended Data Table 3 as follows.

"Unread gap" indicates the number of regions within each RNA-seq contig for which no corresponding sequence was found in the DNA-seq data and whose presence has not been experimentally verified by Sanger sequencing.

- Extended Data Table 4: Definition required for the numbers provided in brackets behind the "Yes" and "Possible" entries under "Phylogenetic analysis" (bootstrap support, I assume(?)).

The caption of Extended Data Table 4 has been revised as follows to include definitions of the terms used.

Extended Data Table 4 | Putative kleptoplast genes compared with the previous transcriptome analysis by Karnkowska *et al.*⁸ and a brief summary of the molecular phylogenetic analysis for each translated peptide. For each *R. viridis* sequence, whether it forms a monophyletic group with Euglenophyceae or *Tetraselmis* spp. is indicated; bootstrap values for such nodes are shown in parentheses (Yes: >90.0; Possible: >75.0). Grouping with other lineages is labeled "No," and cases lacking sufficient bootstrap support are labeled "Poorly resolved." Full phylogenetic trees for each gene product are available at Figshare⁶⁵.

➤ Extended Data Fig. 4: Legend doesn't fit to figure. The described panel a is missing. What are orange and blue in panel b?

The caption of Extended Data Fig. 4 has been revised as follows. Please also note that we re-examined the original primary data obtained by the student co-author, and the plot has been updated accordingly.

Extended Data Fig. 4 | RNA interference (RNAi) experiments targeting *RvRbcS-like*.
a, qPCR measurements showing time-series changes in *RvRbcS-like* expression levels, normalized to those of the α -tubulin gene, of three independent experiments. **b**, Net oxygen evolution rates in wild-type *R. viridis* cells (blue) and RNAi-treated cells (orange) measured on days 8, 12, and 16 after feeding on *Tetraselmis* sp., corresponding to 9, 13, and 17 days after the second electroporation introducing dsRNA into the cells.

Reviewer #3 (Remarks to the Author):

This study by Kashiyama et al. is logical and technically sound. Controls are properly employed, and the results support the main experimental claims. The main contribution of this study is the experimental confirmation of the targeting and translocation of two functionally important kleptoplast-localized nucleus-encoded proteins in *Rapaza viridis*. This is an important technical advance and is much welcomed.

From a biological and physiological perspective, neither the questions nor the results are novel or unexpected. The proteins chosen are expected to be targeted to the kleptoplasts given an abundance of circumstantial evidence provided in previous studies.

For us, the primary purpose of this paper is to introduce *Rapaza viridis* as an experimental organism that anyone can study, one that does not rely on “circumstantial evidence” with respect to the early evolution of organelles. Our previous report (Karnkowska *et al.*, 2023, *PNAS*) was, we believe, novel and important for biological science, but it had major drawbacks that could not be addressed at that time. For example, it did not allow quantitative assessment of gene expression levels from the RNA-seq data.

By generating new quantitative datasets that are independent of the previous report, we now know which genes are truly significant and which are not. In fact, many of the genes turned out to be expressed only at low levels, whereas others are expressed at very high levels. One important, if somewhat cryptic, message of this paper is that *in silico* approaches to poorly understood biological phenomena, when based on limited or superficial data (although often the only option for organisms that are difficult to culture), require very careful interpretation. This was precisely why, in our previous paper, we deliberately restricted our discussion of the cDNA data to horizontal gene transfer and carefully avoided speculating about their actual functions.

At least in our view, there is no “circumstantial evidence provided in previous studies” that could convincingly support the functional significance of such genes, because none of the kleptoplastic organisms or photoendosymbiotic systems studied before *R. viridis* had provided *biochemical*, and therefore conclusive, evidence that endosymbionts or kleptoplasts are molecularly modified by host genes. Although we acknowledge that we ourselves previously provided substantial but still circumstantial evidence for targeting, the present paper offers the *first direct evidence* of actual targeting.

Importantly, the significance of this study goes beyond demonstrating that nucleus-encoded proteins are transported to kleptoplasts and function there, as had been anticipated. Our data further show that the kleptoplast interior is actively remodeled by the host through the formation of a protein complex that differs from its original state, exemplified by the reconstituted RuBisCO complex. This indicates that the kleptoplast is not merely maintained but is molecularly reconfigured by host-derived components.

We are confident that this study will serve as a landmark, moving research beyond the predominantly *in silico*-based cell–cell interaction studies that have accumulated so far and advancing it to biochemical-level analyses that cannot simply be extrapolated from preexisting, minimalistic model organisms.

The study is technically impressive, but the technical advancements (i.e., knock-down, CRISPR-Cas9) had previously been developed and reported by Maruyama et al.

We acknowledge that the technical advancement in genetic manipulation compared with our previous paper (Maruyama *et al.*, 2023, *PCP*) may be minor, but this should not diminish the value of the present manuscript. We believe that this study more effectively highlights the strong appeal of *R. viridis* as an organism that

researchers will be motivated to study through biochemical and molecular-physiological investigations supported by genetic manipulation.

In my opinion, the framing of the manuscript is misleading. The title is rather cryptic and overhyped. These concerns are mostly restricted to the title and the introduction. Unnecessary neologism such as “xenogeneic organelloids” are introduced to presumably exaggerate the importance of this contribution.

Regarding the title, your point may well be justified. In particular, on reflection we felt that the term “organelloid” could be misleading, and we have therefore reverted to “organelle”. What we wished to convey with the original title was that *R. viridis*, which has finally become available as an experimental organism, should not be confined to the narrow context of interest in kleptoplasty alone. As we also state in the Discussion, the importance of being able to study such an experimental organism lies in the possibility of addressing more general scientific questions, such as which combinations of molecular mechanisms present in ancestral organisms underlie innovations like kleptoplasty.

Therefore, we intentionally avoided strongly foregrounding the term and concept of “kleptoplasty” as a well-known phenomenon in the first half of the paper, so that even readers who are unfamiliar with, or not particularly interested in, kleptoplasty would be encouraged to read the manuscript in depth.

The statement in L319-323 is somewhat confusing and not well-informed based on evolutionary theory. Extant living species cannot be considered evolutionary intermediates, but there are characters that can be considered as such. Species express combination of characters that can be considered ancestral or derived. In this way, *R. viridis* is informative as it retains systems that may have been ancestral to photosynthetic euglenids and may therefore inform the evolutionary origin of phototrophy in this group.

First, we apologize for having used wording that could lead to misunderstandings for readers. The intention behind this phrasing was essentially that we did not want *R. viridis* to become widely framed in the public sphere as an organism “on its way to acquiring chloroplasts,” which would, in a sense, put it into a quasi-pseudoscientific context. This concern was partially because this is a broad-audience journal that will also be read by people who are not very familiar with evolution.

We believe we understand your point.

One important issue, in my view, is that we cannot rule out the possibility that many of the elements involved in the kleptoplastic phenomenon of *R. viridis* are in fact “derived” characters. While we do think phagocytosis itself is an ancestral character (rather than something that was re-acquired), we cannot completely exclude the possibility that a cell that once possessed a permanent chloroplast through some *other* process later lost it and transitioned to a kleptoplastic state. Personally, I consider this less likely, but there are so many chloroplast-related genes, including many that are barely expressed, that one is tempted to suspect such a scenario, and it is also puzzling that the *Tetraselmis* nucleus is excluded from the outset. Also, taking this into account, we think we should keep in mind the possibility that, in the future, “kleptoplasty” will no longer be treated as a single homologous phenomenon grouped under one umbrella.

In any case, with the data currently available to us, it is not possible to discuss the common ancestor at a finer resolution, as we also noted in the earlier part of the relevant paragraph. Essentially, we need to wait for the further exploration and discovery of lineages that are sister to *Rapaza* and Euglenophyceae.

Therefore, rather than extending the discussion further here, we have deleted the first sentence of the portion you indicated and revised the text (P14; L317-324) as follows.

All organisms exist in evolutionarily transient states, shaped by continuous evolutionary refinements as they adapt to ever-changing conditions to ensure their survival through successive generations, and *R. viridis* is no exception to this. This species cannot be definitively placed on a trajectory toward acquiring chloroplasts. Nevertheless, *R. viridis* represents a new example of the unexplored ecophysiological potential of eukaryotic cells. Thus, the introduction of *R. viridis* as an experimental model phototroph that may be amenable to genetic engineering, alongside other kleptoplastic phototrophs that have been and will be studied, will expand our understanding of the molecular mechanisms by which eukaryotes interact with the bacterium-derived photosynthetic machinery.

RESPONSES TO REVIEWER COMMENTS

Reviewer #2:

The revised version of the manuscript has improved substantially, and the authors are to be congratulated on the new version. Almost all of this reviewer's previous criticisms have been addressed sufficiently. However, the terms "5% seawater" and "10% seawater" remain unclear and should be specified more precisely as requested previously.

We apologize for the lack of clarity regarding the "5%/10% seawater" conditions. We have now explicitly defined these electroporation conditions in the Methods as 1/20-strength ASW (5% (v/v)) or 1/10-strength ASW (10% (v/v)) supplemented with trehalose at the indicated final concentrations. Importantly, the RNAi electroporation buffer is unchanged from Mariyama *et al.* (2023) (1/20-strength ASW with 500 mM trehalose). In contrast, the CRISPR/Cas9 genome-editing workflow has since been optimized and currently uses a different electroporation buffer (1/10-strength ASW with 450 mM trehalose). We have revised the manuscript to make this method-specific difference explicit and to avoid any impression of a typographical inconsistency.